# Scalable Retrieval of Similar Landscapes in Optical Satellite Imagery Using Unsupervised Representation Learning

Savvas Karatsiolis [1,*], Chirag Padubidri [1] and Andreas Kamilaris [1,2]

1   Cyens Centre of Excellence, Dimarchou Lellou Dimitriadi 23, Nicosia 1016, Cyprus; c.padubidri@cyens.org.cy (C.P.); a.kamilaris@cyens.org.cy (A.K.)
2   Department of Computer Science, University of Twente, 7522 NB Enschede, The Netherlands
*   Correspondence: s.karatsiolis@cyens.org.cy

**Abstract:** Global Earth observation is becoming increasingly important in understanding and addressing critical aspects of life on our planet, including environmental issues, natural disasters, sustainable development, and others. Finding similarities in landscapes may provide useful information regarding applying contiguous policies, by making similar decisions or learning from best practices for events and occurrences that previously occurred in similar landscapes in the past. However, current applications of similar landscape retrieval are limited by a moderate performance and the need for time-consuming and costly annotations. We propose splitting the similar landscape retrieval task into a set of smaller tasks that aim to identify individual concepts inherent to satellite images. Our approach relies on several models trained using unsupervised representation learning on Google Earth images to identify these concepts. We show the efficacy of matching individual concepts for retrieving landscape(s) similar to a user-selected satellite image of the geographical territory of the Republic of Cyprus. Our results demonstrate the benefits of breaking up the landscape similarity task into individual concepts closely related to remote sensing, instead of applying a single model targeting all underlying concepts.

**Keywords:** Earth observation; landscape similarity; image retrieval; satellite images; unsupervised learning

## 1. Introduction

A growing amount of remotely sensed images and environmental data retrieved from in situ sensors are being made available to scientists, policymakers, and stakeholders, allowing them to make critical decisions about important aspects of the planet's future (e.g., environmental policies) [1]. This growing accumulation of data raises the need to better process and exploit these abundant big data [2]. Satellite imagery is becoming increasingly available at unprecedented spatial and temporal resolution, allowing more sophisticated tasks to be tackled and more interesting analyses to be conducted [3]. Meanwhile, the environmental disturbance and the endangerment of ecosystems caused by human actions magnify the need to better understand, process, and analyze landscapes [4,5]. Analyzing landscapes and identifying similar ones can support better policy making and decision making and may reveal relations or features between landscapes located far away from each other that were not realized before. For example, retrieving landscapes like a user-selected landscape may provide useful information regarding applying similar policies or taking contiguous measures against events that have already occurred in the initial landscape e.g., wildfires, contagious diseases, earthquakes, flooding, war-related actions, pollution, and contamination, etc. Likewise, other predicted events or associations may relate to imminent overcrowding, near-future land cover or land use change of some kind, or even certain deterioration risks like vegetation wilting or water stress to crops due to prolonged heat and drought [6,7]. Besides the similarities and associations based on environmental

aspects, landscape similarity could also be used to associate geographical areas in various contexts. For example, an image retrieval engine could associate landscapes based on economic-related factors like land price inflation/deflation tendency, property estimation and risk/vulnerability analysis, making it a useful tool for real estate professionals and investors, as well as insurance companies.

Early works in quantifying landscape similarity in the optical remote sensing field relied on landscape metrics, and much of the work in this direction was conducted in the context of land use/land change (LULC) research [8–11]. Initially, partly because the resolution of available satellite images was very low and since feature extraction techniques based on advanced image processing and computer vision methods such as deep learning (DL) had yet to be developed and refined, several works [12–14] used distribution statistics (first- and second-order statistical summaries) as landscape similarity metrics (LMs). These simple LMs proved to be effective when Euclidean distance was used to compute the similarity between two landscapes. Such approaches led to the development of FRAGSTATS [15], a very popular spatial pattern analysis program for categorical mapping. Besides FRAGSTATS, a lot of contemporary spatial analysis software uses similar metrics. Despite their effectiveness and ease of use on low-resolution satellite imagery, first- and second-order landscape statistics are not suitable for high-resolution imagery because they do not capture the semantics of the image and do not comprise a robust methodology for identifying high-level image/landscape similarities. Image statistics constitute a signature that may be used to identify similar depictions in a more general sense, e.g., the depiction of a pasture, an urban area, an industrial area, etc. However, such simple statistics cannot go beyond very distinct image features. Image-statistic-based LMs can be used in various formulations to construct a representation vector of a landscape that is then used to compute the similarity measure with other landscapes. An assessment of the performance of such different formulations was made by Niesterowicz and Stepinski [16].

In recent years, in the era of big data and DL, image retrieval has focused on the extraction of features from deep convolutional layers. Shi et al. [17] proposed a simple but effective method called strong-response-stack-contribution (SRSC) that facilitates the construction of better representations by focusing on a suitable region of interest (ROI). Similarly, Chen et al. [18] proposed the identification of an ROI in the image, which is then used to extract a set of features from the fully connected layer of the model. Another interesting approach by Babenko et al. [19] proposed aggregating the local features to form new powerful global descriptors. Finally, a survey on DL-driven remote sensing image scene understanding, including scene retrieval, was published by Gu et al. [20].

In this work, we present a proof-of-concept application of landscape similarity matching with unsupervised representation learning (URL). Following the DL paradigm, we learn the features of landscapes depicted in Google Earth satellite images, using the island of Cyprus as our case study [21], by training a model in an unsupervised manner (without using any annotations/labels) and then identifying the most similar landscapes to a user query by computing representation distances. Our approach refrains from using any labels by taking advantage of the knowledge gained from learning to identify self-sustained concepts that are essential to remote sensing imagery, e.g., the road network, the buildings, and the tree configuration, or particularly notable or unique characteristics regarding the depicted surfaces, textures, or shapes. Learning the arrangement of these objects requires the use of additional DL models that extract the concept information from the satellite images. However, once these models are trained, developing a similarity model is achieved with URL, without the need for expensive annotations. A similar work by Aksoy et al. [22] used a large-scale benchmark archive called BigEarthNet [23], consisting of Sentinel-1 and Sentinel-2 satellite images from 10 European countries and a web application (EarthQube) for image retrieval. EarthQube uses a deep hashing algorithm named metric-learning-based deep hashing network (MiLaN) [24,25] to implement similar image retrieval. MiLaN was trained with the triplet loss function [26,27] to learn a metric space where similar images are mapped close to each other, while dissimilar images are mapped far from each other. While

MiLan is effective in retrieving similar landscapes, it requires extremely heavy labeling, since the images must be annotated with multiple labels to facilitate the triplet loss in calculating an appropriate metric space.

We argue that annotating images for the landscape similarity problem is by nature an inaccurate process, because a human labeler must consider and carefully assess every candidate landscape before assigning an accurate similarity score to image pairs. This becomes infeasible when studying large geographical areas. To delineate the landscape similarity scale and correctly quantize it, a comprehensive study of every image in the dataset is required, which is extremely difficult and time consuming. Often, to resolve this problem, researchers compromise by using less accurate labels or using binary labels that are much cheaper to obtain and may be used to train the model with triplet loss [24,25]. Even in the case of using suboptimal labels such as binary labels with triplet loss, assigning good image associations requires studying the dataset and choosing the most eminent associations among a huge number of possibilities. Being notoriously difficult and expensive to annotate, landscape similarity tasks tend to benefit less than expected from supervised learning in terms of performance and development cost. Thus, we believe that landscape similarity inferring models like the one we propose, which are trained in an unsupervised manner and do not require an annotation process, may achieve at least the same performance as supervised methods, with a much cheaper development and maintenance cost.

In contrast to related work, the contribution of our work is the proposal of a new method for landscape similarity matching from optical satellite imagery based on the URL technique and which does not require heavy data annotation. A key finding, discussed and demonstrated in Section 3, is the efficacy of breaking up the landscape similarity task into individual concepts closely related to remote sensing, instead of trying to capture all concepts and image semantics with a single model like a single RGB semantics model.

## 2. Materials and Methods

Defining what makes two optical satellite images similar or not is not a precise scientific task. There are several aspects of satellite images that comprise what humans perceive as similarities between two satellite images of different landscapes, e.g., the colors, the shapes of the buildings, the tree configuration, the texture of the surfaces, the high-level semantics of the depicted landscape, etc. The most popular similarity metrics used for image retrieval rely on either identifying common structures in the images (e.g., based on trivial image statistics or LMs) or comparing the query image's embedding against the embeddings of the images stored in a database (based on distance calculation metrics). While the former approach usually matches images based on sparse but highly distinct similarities such as textures and common structural components, it often fails to capture whether two images are similar in the broader sense, e.g., they share the same high-level semantics. Likewise, image embeddings tend to match broader concepts encapsulated in a high-dimensional vector but may ignore sparse but distinct features shared by two images, especially when the images do not have many high-level concepts in common. In addition, image similarity depends on the content of the images and the image domain: when quantifying the similarity between two images, we tend to match different domain aspects for closed-caption photography than for satellite imagery.

To face the challenge of measuring the similarity between two satellite images, we propose to divide the similarity identification problem into four smaller (sub)tasks that are closely related to remotely sensed optical imagery:

a. the task of finding landscapes with a similar road network;
b. the task of finding landscapes containing a similar building configuration;
c. the task of matching the tree configuration in the query image;
d. the task of matching the high-level semantics of the query image.

The first three concepts (roads, buildings, trees) comprise dominant components of satellite images that occupy a huge portion of the image semantics and monopolize the

observer's interest. The high-level semantic-matching task complements the operation of the dominant components and is particularly useful when the landscape contains no or very few roads, buildings, or trees. By breaking the task into smaller subtasks, it is also possible to apply weighting to each task and thus control its influence on the outcome. For example, a user might be more interested in the similarity of the road network instead of the tree arrangement in an area. In this case, the calculated road network similarity should receive a higher weight than the tree arrangement similarity. Likewise, another user might only consider two of the applicable similarity aspects and ignore the remaining one. The details of applying weights to the similarity concepts are presented in Section 2.3.

The following subsections present the DL models and methodologies used for learning appropriate representations to implement the similarity metrics, explain the embedding database details, and describe the image retrieval process.

### 2.1. Similarity Models and Algorithms

Our approach uses URL to discover the underlying data clusters, apply data groupings, and identify hidden pattern associations, without requiring labels of any kind. In contrast to supervised learning (SL), where the model is trained on a set of annotated/labeled data, URL aims to discover patterns and structure in unlabeled data, clustering similar data points together and extracting data features in an unsupervised manner. URL has numerous applications in computer vision and natural language processing, especially in AI applications for which data labeling is costly and time-consuming. We use four different DL models trained with URL, one for each similarity aspect we explore (roads, buildings, trees, and general image semantics).

### 2.1.1. Unsupervised Representation Learning and the Barlow Twins Algorithm

Many algorithms implementing URL are based on contrastive learning (CL) [28], which is about training a model to distinguish between pairs of similar and dissimilar patterns. Simple contrastive learning of representations (SimCLR) [29] is one of the most popular URL algorithms and uses different views of the same data points to map similar inputs to nearby points in the feature space and dissimilar inputs to distant points. The technique of using different views of the same data with contrastive learning is used by many URL algorithms besides SimCLR, like momentum contrast (MoCo) [30], bootstrap your own latent (BYOL) [31], Barlow twins model [32], and the simple Siamese model (SimSiam) [33]. MoCo incorporates a momentum-based update that smoothens the representation space and improves training stability. BYOL trains a model to predict the features of a different view of a data point given another different view of the same data point. The Barlow twins (BT) model also uses different views of the same data points to reduce the correlation (redundancy) between the different dimensions of the representations. To reduce the redundancy between the learned representations, the BT model computes the cross-correlation matrix $\mathbb{C} \in \mathrm{R}^{m \times m}$ of the output features along the batch dimension of size $m$. This strategy makes the model learn meaningful representations that contain discriminative domain features. The loss function of the model is defined as

$$\mathcal{L}_{\mathcal{BT}} = \sum_{i} \left(1 - \mathbb{C}_{ii}\right)^2 + \lambda \sum_{i} \sum_{i \neq j} \mathbb{C}_{ij}^2 \tag{1}$$

The first term of the loss is the invariance term, which tries to equate the diagonal terms of the cross-correlation matrix to 1 and thus make the embeddings invariant to the augmentations applied. The second term is the redundancy reduction term that pushes the off-diagonal terms of the matrix towards 0 and thus decorrelates the embeddings of non-similar images. The hyper-parameter $\lambda$ controls the effect of the second term on the overall loss function. By generating normalized representations, the Barlow twins model allows direct application of the cosine similarity metric to the embeddings and thus the derivation of an intuitive metric for identifying patterns of similar semantic content.

The BT algorithm, like every other URL algorithm, uses augmentations to produce different views of data points. URL algorithms often use the same or a very similar augmentation queue to the one proposed by [29] to create different views for every data point: color jitter (brightness, contrast, saturation, and hue), random gray scaling, resizing/cropping, and random horizontal flipping. In particular, the BT algorithm achieves state-of-the-art results on image classification benchmarks and, very importantly, it is not as sensitive to the batch size used during training as other URL methods. The BT algorithm is also less sensitive to the number of negative examples required for each data point to obtain satisfactory results, in contrast to other methods. These advantages also contribute to more stable training and render the BT model suitable for our case, because of the memory constraints imposed by using large-sized satellite patches in the training set. Specifically, we use satellite patches of size $512 \times 512 \times 3$, depicting a reasonable area of a landscape and thus capturing an adequate portion of the region's semantics. Such a large patch size limits the use of large training batches due to memory constraints, making the BT algorithm a good fit for the current study.

2.1.2. Using Unsupervised Representation Learning to Develop Similarity Models

We used four ResNet34 [34] models trained with URL on hundreds of thousands of Google Earth images of Cyprus. During inference, we considered the cosine distance between the embedding of the query image (the landscape selected by the user) and the embeddings of every image in the database, and the top-$K$ images with the smallest distance to the user query were returned.

We used Google Earth images for our image retrieval application because they provide high spatial resolution (~30 cm/pixel), they are relatively easy to acquire, and they are (fairly) frequently updated (i.e., every 4–6 months). The satellite images used in this study were acquired on June 2022 and spanned the southern part of the island of Cyprus. The images were split into patches of $512 \times 512$ pixels, providing nearly 4 million patches in total. We used 80% of the patches for training the URL models with the BT algorithm and 20% of the patches for test purposes (visual inspection of the similarity between retrieved images during inference). In all four models, the backbone was a ResNet34 architecture outputting a multidimensional normalized representation for every input satellite patch.

As described in Section 2.1.1, the BT algorithm was used for training models to output similar embeddings for similar inputs and dissimilar embeddings for dissimilar inputs, according to a certain concept each time. As mentioned above, we used four concepts that are dominant in remotely sensed optical images: road networks, building arrangement, tree arrangement, and general image semantics. Each model built a feature space for the concept it was exposed to and learned how to combine the emerging domain patterns to calculate the embedding of an input. Since the models' training objective function minimizes the cosine similarity between the embeddings of similar images, the model learned to map images of similar appearance to feature space areas that were in proximity in terms of cosine similarity. However, examining the overall similarity of two images by considering various aspects of them required some modifications to the way the original BT algorithm creates different views of the data points (see Section 2.1.3). For example, the road similarity model should receive as input a binary mask representing the road network depicted in the image. The building similarity model should receive as input a binary mask showing the buildings in the image, while the tree similarity model should receive as input a binary mask representing the trees in the image. The semantic similarity model accepts RGB images as input and complements the similarity models that perform inference based on binary masks of the main concepts (roads, buildings, trees). The road mask was extracted from a Geographic Information System (GIS) road network layer of the island of Cyprus. The buildings' mask was extracted from a GIS layer created by the authors with an AI segmentation model they developed that identifies the contours of buildings depicted in Google Earth images [35]. Finally, the tree mask was extracted from a GIS layer created by the authors with an AI tree detection model they developed to count trees in satellite

images [36,37]. Figure 1 shows an example of a Google Earth image and its concept masks extracted using the GIS layers empowered by the AI models mentioned above. Figure 2 shows the data flow of the proposed approach for landscape similarity matching.

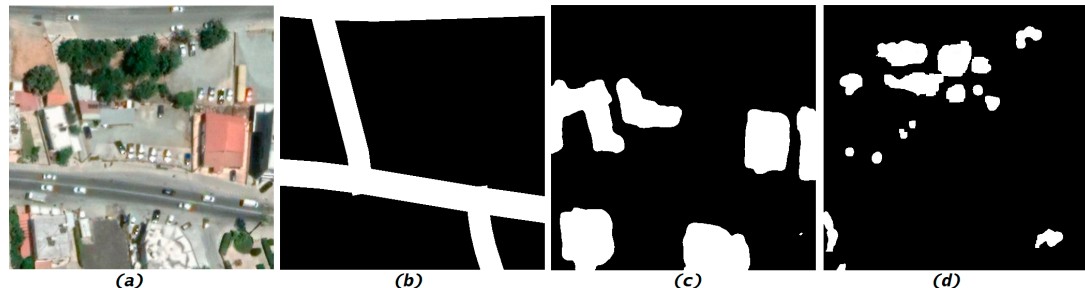

**Figure 1.** To determine the similarity between two satellite images, we examine three major concepts that are paramount in landscape similarity determination: the road network, the building surfaces, and the tree arrangement. The building and tree maps were obtained from proprietary AI models developed by the authors and the road network was obtained from a GIS layer. From left to right: (**a**) the original RGB satellite image, (**b**) the binary mask of the road network, (**c**) the binary mask of the building surfaces, and (**d**) the binary mask of the tree foliage.

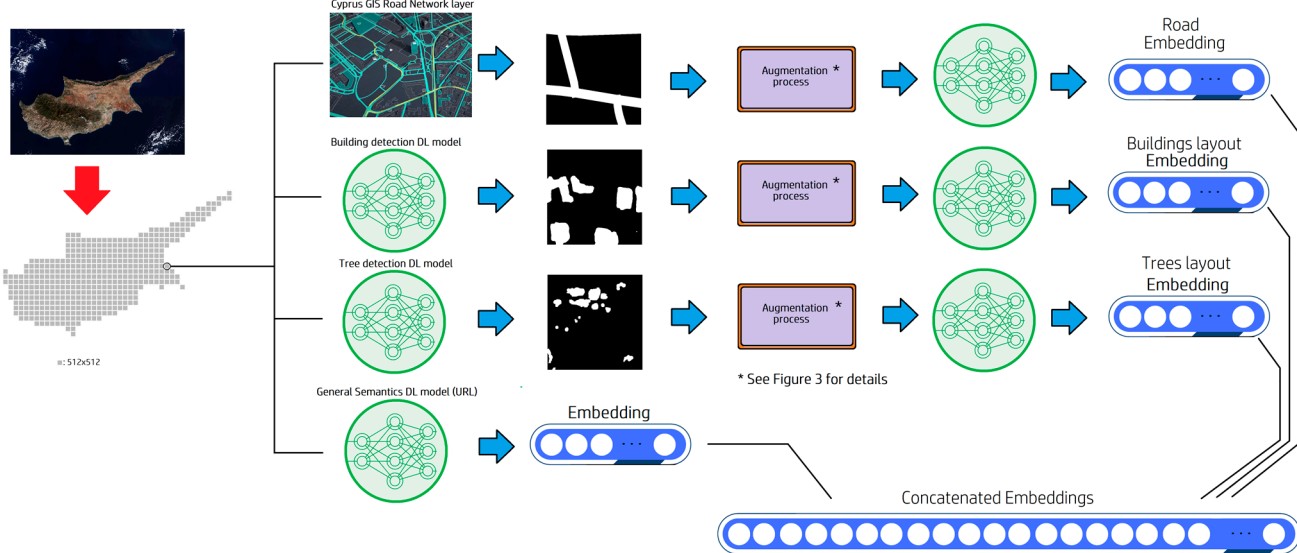

**Figure 2.** The data flow of the proposed landscape similarity identification approach. A GIS layer provides the road network binary map, building and tree detection DL models provide the tree and building masks, while another DL model extracts an embedding for the RGB semantics. Each binary mask type (roads, buildings, trees) goes through an augmentation process to train a similarity model using URL. The embeddings calculated by the similarity models are normalized and then concatenated with the normalized RGB embedding to form the final embedding of each satellite patch.

When assessing whether two images are similar or not (or calculating the similarity between two images) our approach examines the three concepts' masks in pairs and computes a similarity score for each pair. In addition to these scores, our approach also computes a similarity score for the two RGB images and combines all four scores into a final similarity measure. We note that, in practice, our approach does not use the developed AI detection models (building and tree detection AI models) for inferring the binary maps of each patch, because this would be very costly. Instead, the AI detection models were used to create two GIS layers, one for buildings and one for trees, which are used for fast extraction of the binary masks each time a new patch is processed. Figure 3 shows the

process of inferring the similarity score of two concept masks and the training process of the model. During training, two masks are created from the same image, one being the actual concept mask of the image and the other being a modified version of it. Retrieving the most similar image to a certain selected image (the query image) requires computing the similarity score between the query and all the images in the database and then selecting the image in the database that has the smallest distance to the query image. This procedure can be very inefficient and time-consuming if not implemented correctly and is discussed in detail in Section 2.2.

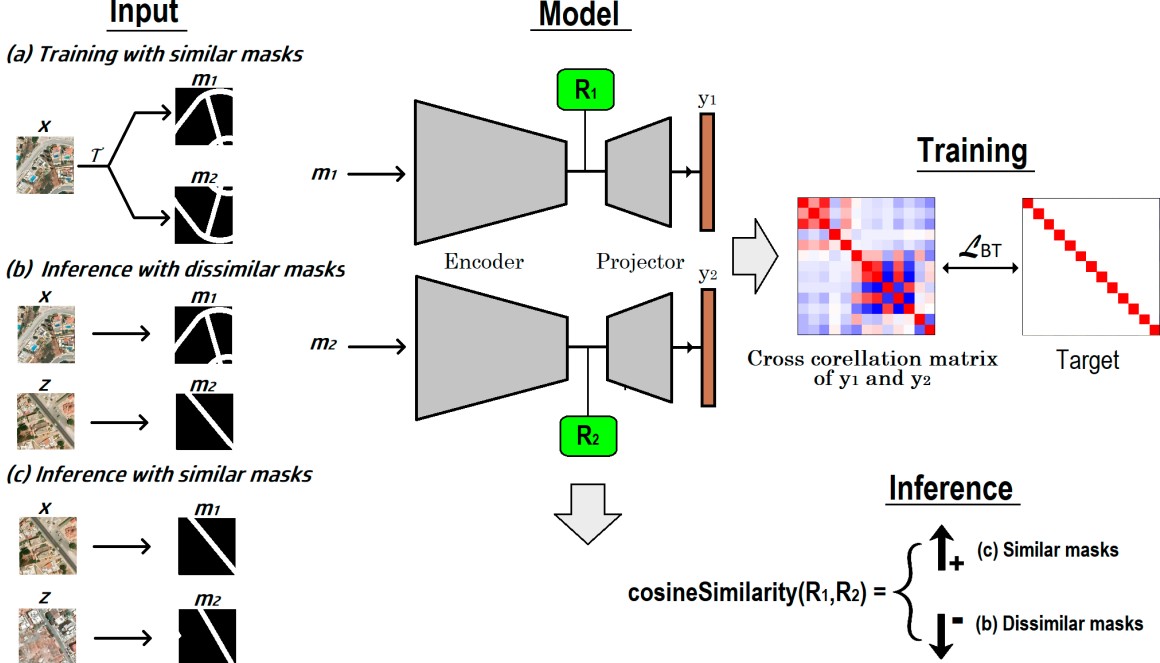

**Figure 3.** Training and inferring concept similarity based on the Barlow twins algorithm. The example relates to the road network concept, but the processes generalize to other similarity concepts (buildings, trees). (**a**) Training with similar masks: Two views of image $x$ ($m_1$ and $m_2$) pass through the model whose weights are adapted so that the cross−correlation matrix of the two output embeddings $y_1$, $y_2$ has a diagonal with values *1* and every off−diagonal element is zero. (**b**,**c**) Inference: The concept masks extracted from two different images $m_1$ and $m_2$ pass through the model to obtain the intermediate representations $R_1$, $R_2$ and their cosine similarity is computed. If the images are similar, the cosine similarity should be high, otherwise it should be low.

In the following sections, we present the four models of our approach, one for each similarity concept considered and described above. We also discuss how the models are trained, i.e., how the different concepts' views are created for each model and how they are used to train the models.

### 2.1.3. Creating the Training Data for the Road Network Similarity Model

We applied the BT algorithm to train a ResNet34 model to output similar embeddings for alike road network masks. Specifically, the embeddings of two similar road network masks should be close to each other, while the embeddings of two dissimilar road network masks should be far from each other.

The conventional augmentations used by SimCLR, the BT algorithm, and many other URL algorithms are not suitable in our case, because the input to the model is a binary mask and not an RGB image. Having a binary mask as input to the model renders the color jitter augmentations (brightness, contrast, saturation, and hue) useless. Furthermore, since we aimed to train a model on a concept sourced from the entirety of the input image and not from certain regions only, we also did not use resizing and cropping. We created pairs

of similar road network masks by modifying a certain binary mask with morphological operations, translation, and rotations of multiples of 90°. First, the mask goes through 10 dilation/erosion steps of random intensity and kernel sizes. The dilation steps enlarge and expand the active areas of the mask, making them wider, while erosion reduces the size of the active mask areas and gradually diminishes them. Then, the mask is horizontally translated (shifted) in either direction by a random number of pixels (−20 to +20 pixels). Finally, the mask is rotated by an angle of 90°, 180°, or 270°, with a probability of 0.3 for each rotating angle (the probability of not rotating the mask is 0.1). The pseudocode for creating pairs of similar road network masks to train the model is shown in Table A1 in Appendix A.

### 2.1.4. Creating the Training Data for the Building Layout Similarity Model

Like the road network model (and the models of all the concepts we examine), the building layout similarity model is a ResNet34 network trained with the BT algorithm. The data used to train the building layout similarity model were generated using an augmentation scheme tailored to the features of the building layout masks. As shown in Table A2 (Appendix A), we create bounding boxes containing the buildings in a mask (each bounding box encloses one building) and then we randomly translate each of the bounding boxes by ±20 pixels. Then, the resulting mask is modified via 10 steps of random erosion/dilation operations, with kernels of random size. Finally, the mask is randomly rotated by an angle of 0°, 90°, 180°, or 270°. In contrast to the road network model augmentation process, the no-rotate probability (0° rotation) is equal to the probability of rotating by any other angle. This strategy gives better results, because the building masks inherently have more variation than the road masks and we can compensate for the road masks' lower variance by applying more rotations.

### 2.1.5. Creating the Training Data for the Tree Arrangement Similarity Model

The augmentation process used in feeding the tree arrangement similarity model with similar pairs of masks was the same as the augmentation process of the building layout similarity model (Table A2, Appendix A), with the only difference being the use of a random translation of ±15 pixels instead of ±20 pixels. Tree masks usually contain many more objects than building masks, and thus they inherently possess a higher variation, canceling out the need to vary them significantly.

### 2.1.6. Creating the Training Data for the RGB Similarity Model

The RGB similarity model captures more general similarities than the ones captured by the models that process binary masks of certain concepts. Since the model's input is an RGB image, the augmentation process for generating similar image pairs followed the same conventional augmentation queue as proposed by SimCLR, creating different views of a certain RGB image.

In this case, the augmentation queue is shown in Table A3 (Appendix A) and includes a resize/crop augmentation, followed by a left-to-right flip, color jitter augmentation, and finally a color-dropping step.

### 2.1.7. Implementation details

We used a training batch size of 256 and mixed floating point (FP16) precision to reduce the memory requirements and speed up the training. For all four similarity concepts, we used an embedding size of 256. To train the model, we used eight RTX A5000 GPUs (Nvidia (Santa Clara, CA, USA)) and an Adam [38] optimizer with a fixed learning rate of $5 \times 10^{-4}$ and a weight decay of $1 \times 10^{-6}$.

### 2.2. Similar Landscape Retrieval Process

The image retrieval process relies on the embeddings provided by the similarity models. Assuming a region of interest (e.g., the area of a city, a country, or a broader

region on Earth), the satellite imagery acquisition from this region is kept in storage and the details (file path and geographical coordinates) are stored in a MySQL database [39] for fast information retrieval. In addition, the embedding of each satellite image is precomputed and stored in a Milvus vector database [40]. When we want to match a place found in the region of interest to other places in the region, we compare the embedding of the query satellite image with the embeddings stored in the vector database. The entries with the most similar embeddings are returned, while their coordinates and info about the file paths of the corresponding satellite images are retrieved from the MySQL database.

The use of a vector database is crucial for making the solution scalable. Even in case studies covering small regions of interest like this study, in which we only focus on the southern part of the island of Cyprus, the number of satellite images is large ($\sim 4 \times 10^6$) and can easily explode to an unmanageable size when considering a large country, a continent, or even the whole planet. Refraining from using a vector database and finding the most similar place in the region by calculating the similarity of a query through comparing with every image in the database, imposes a limit on the size of the studied region. This problem emanates from the accumulating computational cost of calculating the similarity between the query image and every image in the database in a naive way (applying the similarity formula sequentially for all possible embeddings and then choosing the one with the closest value). This simple image retrieval process has a complexity of $\Omega(N)$, where $N$ is the number of images in the database, which makes it very inefficient for huge $N$. Precomputing the cosine similarity of every image in the database with every other image in the database and thus making the similarity measures readily available during inference is also impractical for a substantial number of images, because the complexity of such an approach is $\Omega(N^2)$. Furthermore, for each future image added to the database, $\Omega(N)$ similarity computations are needed, which reduces the scalability of the "precomputed similarities" strategy.

### 2.2.1. Vector Databases

Vector databases offer optimized storage and querying capabilities for embeddings, in contrast to traditional scalar-based databases. Working with vector embeddings imposes challenges on real-time analysis, scalability, and performance that traditional scalar-based databases cannot address properly. High-dimensional indexing databases add significant capabilities to modern AI systems, such as information retrieval and long-term memory, and play a crucial role in managing and retrieving information from large datasets with complex structures, enabling applications that require efficient similarity-based querying and analysis. Vector databases commonly use methods like tree-based structures [41], graph-based indexing [42], and hashing techniques [43,44] to organize and search data efficiently. They often support distance metrics such as Euclidean distance, cosine similarity, Jaccard similarity [45], and more, allowing users to define the notion of similarity that best suits their data domain.

Milvus is an open-source vector database built for similarity search and analytics. It supports a variety of distance metrics and indexing methods. It is particularly well-suited for applications involving machine learning, computer vision, natural language processing, and other tasks/challenges that deal with complex data types represented as vectors. Milvus is designed to scale horizontally, meaning it can handle growing datasets and increasing query loads by distributing data across multiple nodes or servers, which makes it suitable for large-scale applications. Furthermore, it addresses the challenges of efficiently managing and querying high-dimensional data, providing developers and researchers with a powerful tool for building applications that rely on similarity search and analysis.

### 2.2.2. The Landscape Retrieval Pipeline

The landscape retrieval system is built around three main components: the map service, the MySQL database, and the Milvus vector database. As described above, all

satellite images (512 × 512 patches) of the ROI are kept in storage and their information is maintained in a table in the MySQL database. The Milvus vector database stores the embeddings of all satellite patches using the same IDs as in the MySQL database and performs the embedding similarity-matching operation. Figure 4 shows the image retrieval pipeline, the components of the system, and their interaction.

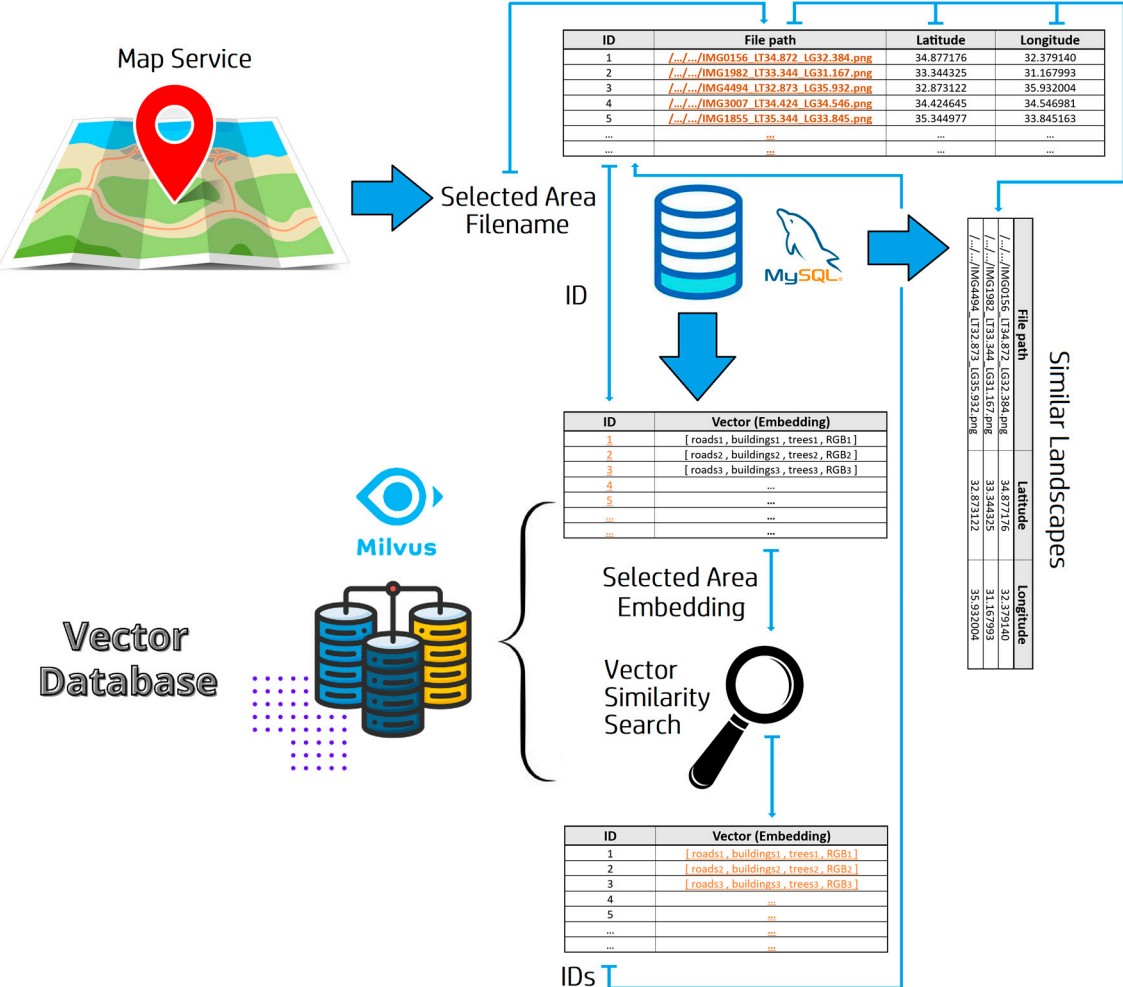

**Figure 4.** The retrieval pipeline: A map service handles the user's query and returns the information of the user-selected area, which is used to retrieve the area ID from the MySQL database. The ID of the query image is used to retrieve the query image embedding from the Milvus vector database, which is then used in the similarity search. The similarity search returns the ID(s) of the most similar landscape(s) in the vector database. Finally, the information associated with the ID(s) returned by the similarity search is retrieved from the MySQL database.

The identification of similar landscapes based on a query image starts with the interaction with an online map service that allows the user to select a certain location of her choice contained in an ROI. Next, the map service translates the selected coordinates into a file path, which is used to query the MySQL database to obtain the ID of the query image. Then, this ID is used to retrieve the embedding of the query image from the vector database. After the query embedding is available, an embedding similarity search is conducted to find the ID(s) of the image(s) with the most similar embedding(s). Finally, the retrieved ID(s) is/are matched in the MySQL database and the details of the results (file path, coordinates) are returned to the user.

*2.3. Tuning the Importance of the Similarity Aspects*

As shown in Figure 2, the similarity aspect embeddings (road network, building layout, tree arrangement, and the general RGB semantics) are concatenated into a single embedding that holds all the representations of the individual aspects together. The individual embeddings are normalized, and thus calculating the dot product between two embeddings is equivalent to calculating the cosine similarity between them. Assuming the similarity aspect embeddings $\vec{R} = [r_0, r_1, r_2, \ldots r_N]$, $\vec{B} = [b_0, b_1, b_2 \ldots b_N]$, $\vec{T} = [t_1, t_2, t_3 \ldots t_N]$, and $\vec{S} = [s_1, s_2, s_3 \ldots s_N]$ for the road network, the building layout, the tree arrangement, and the general RGB semantics, respectively, and with $N$ being the size of the embeddings, then the concatenated embeddings are of the form $[\vec{R}, \vec{B}, \vec{T}, \vec{S}] = [R_0, R_1, R_2, \ldots, R_N, B_0, B_1, B_2, \ldots, B_N, T_1, T_2, T_3, \ldots, T_N, S_1, S_2, S_3 \ldots S_N]$. Since all embeddings are normalized, it holds that $\|\vec{R}\| = \|\vec{B}\| = \|\vec{T}\| = \|\vec{S}\| = 1$. To make a certain similarity concept more important during a similarity search, we only need to multiply the embedding of a certain concept with a scalar $a > 1$. This is true because of the homogeneity property of the norm of a vector $\vec{x}$, i.e., $\|a\,\vec{x}\| = |a|\,\|\vec{x}\|$. Since we want to keep the stored embeddings (in the vector database) unmodified (normalized) and apply similarity-concept weighting of any magnitude at any time, we chose to apply the importance scaling to the query embeddings $\vec{Qr}$, $\vec{Qb}$, $\vec{Qt}$, or $\vec{Qs}$, each corresponding to one of the similarity concepts. Of course, one can apply an importance scaling to more than one query embedding at the same time. A similarity measurement between a query landscape embedding $[\vec{Qr}, \vec{Qb}, \vec{Qt}, \vec{Qs}]$ and a candidate entry in the database $[\vec{R}, \vec{B}, \vec{T}, \vec{S}]$ is as follows:

$$DP = Qr_0.R_0 + Qr_1.R_1 + \ldots + Qb_0.B_0 + Qb_1.B_1 + \ldots + Qt_0.T_0 + Qt_1.T_1 + \ldots + Qs_0.S_0 + Qs_1.S_1 \qquad (2)$$

with DP representing the dot product operation. Using the importance scaling scheme and assuming four scaling factors for the query embeddings, such as $\overrightarrow{Qr_{scaled}} = a.\vec{Qr}$, $\overrightarrow{Qb_{scaled}} = \beta.\vec{Qb}$, $\overrightarrow{Qt_{scaled}} = \gamma.\vec{Qt}$, and $\overrightarrow{Qs_{scaled}} = \delta.\vec{Qs}$, the dot product (DP) of the similarity check comprises four terms: $DP = \alpha.DP_r + \beta.DP_b + \gamma.DP_t + \delta.DP_s$ with each term holding the dot product of one of the four concepts. This result shows that each aspect contributes to the overall result with a quantity that is scaled by the weight of the specific concept embedding. Controlling the significance of each of the concepts examined is very useful especially when the query landscape is complex and the similarity between itself and the candidate landscapes is not immediately evident and is thus difficult to quantify.

## 3. Results

We applied the proposed landscape similarity algorithm to retrieve landscapes similar to areas located in the southern part of the island of Cyprus, which was used as our case study. To avoid the trivial case of getting nearby or neighboring coordinates as a result, we increased the number of candidate landscapes returned by the similarity search and chose the first one in the returned similarity ranking that was located at least one kilometer away from the query landscape. We used the simplest experimental setup by assigning the same importance to all similarity aspects, i.e., no scaling was applied to the query embeddings. We followed the pipeline described in Figure 4, i.e., choosing a landscape from a Google Earth region and using its coordinates to obtain the file path of the saved image via a dictionary created during the training of the models. The map and the dictionary that links the coordinates of a landscape to the storage location of its satellite image comprise the map service shown in Figure 4; in a commercial application, the map service could be a web service incorporating the map and the mapping dictionary. The file path is used in a MySQL query that retrieves the ID of the landscape the user selects. This ID is also kept in the vector database and is thus used to retrieve the embedding of the query landscape. Then, the vector database searches for the *k*-most similar landscapes to the query embedding,

and the first entry from the ranked results whose coordinates are at least one kilometer away from the query's coordinates is selected. In our experiments, we used $k = 10$. Figure 5 shows the results of several landscape similarity searches, picking the best match from the top-10 matches retrieved. From the examples visualized in Figure 5, the reader can observe that the similarity decision was affected by the shape of the road network, the arrangement of the trees, and the shapes and placement of the buildings in the query image. The results suggest that the algorithm considered all the similarity concepts and returned an outcome that constitutes a reasonable compromise between them: even when there was a much better match for each of the concepts separately, the algorithm returned a landscape that generally satisfied the specifications imposed by all binary masks and the RGB semantic model. This is demonstrated even more clearly in Figure 6, which shows the best matches per individual concept inferred using the cosine similarity of each concept embedding individually, instead of the concatenated concept that is illustrated in Figure 2. The returned landscape computed by considering the concatenated embeddings' vector corresponds to neither of the individual masks. On the contrary, the returned landscape concept masks are on par (in terms of similarity) with the masks of the best matches in a way that balances the overall similarity between the individual concepts to achieve a reasonable compromise driven by the cosine similarity metric. An effective way to perturb the consistency of returning a landscape that balances the matching of the individual concepts is to apply a weight to the concept embeddings, as described in Section 2.3. By doing so, the returned landscape focuses more on a certain concept, depending on the magnitude of the weight applied to the specific concept. The results also suggest that the algorithm returned landscapes of similar classes, i.e., an urban landscape was returned when the query landscape was urban, and a rural landscape was returned when the query landscape was rural. This consistency was evident throughout our experiments.

The results also demonstrated the efficacy of breaking up the landscape similarity task into individual concepts closely related to remote sensing, instead of trying to capture all concepts and image semantics with a single model like a single RGB semantics model. Using only one such model and refraining from using the proposed individual concept-matching approach does not capture the essence of landscape similarity, despite single models trained with URL algorithms being effective with closed-caption imagery containing objects, persons, and sceneries at smaller scales, e.g., with datasets like the ImageNet [46]. A model trained with URL on RGB images captures useful content structures, semantics, and texture/color characteristics, which is impossible to achieve with binary masks alone. Specifically, the use of the RGB semantics model in the proposed approach is essential for capturing the characteristics of large surfaces in optical satellite imagery, which adds a particularly useful dimension for the problem of identifying landscape similarity. However, while the RGB semantics model provides useful insights, it is not sufficient on its own to tackle the problem. The RGB semantic model captured the textures and colors of the query image but neglected the arrangement of the objects captured by the models trained with binary masks. Throughout our experimentation, the returned landscape was rarely the same as the best match of the RGB semantic model, but its textures and colors were heavily determined by it. This behavior is highly desirable because the individual concepts complement each other and act synergistically to identify the best match for a certain query. Finally, these results suggest that a multi-concept approach is a better fit than a single-model (single-concept) approach for identifying landscape similarity.

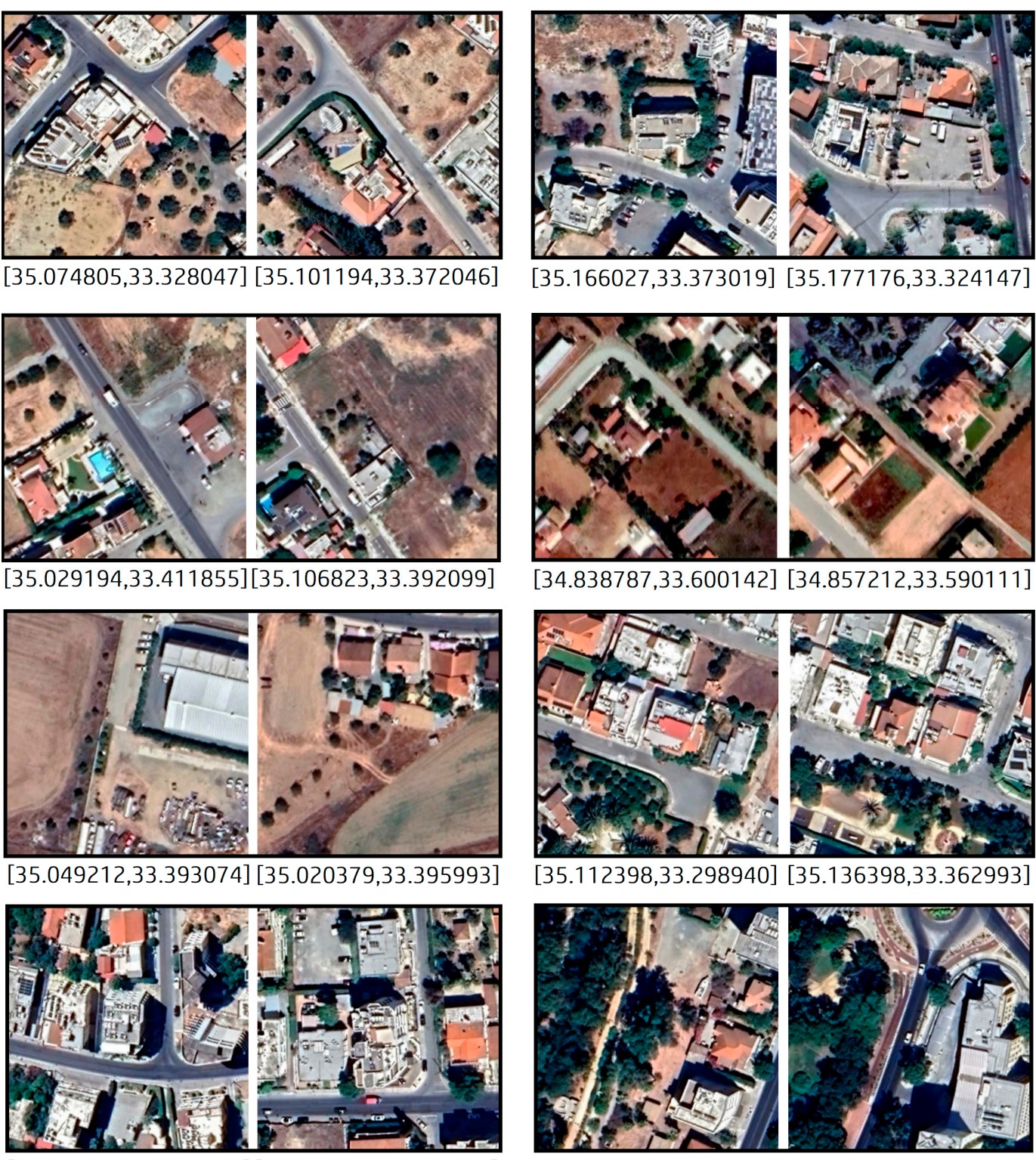

**Figure 5.** The results of several landscape similarity searches. In each pair, the left satellite image is the query image and the right image is the output of the similarity search algorithm. The coordinates (longitude, latitude) of each satellite image are shown below the images.

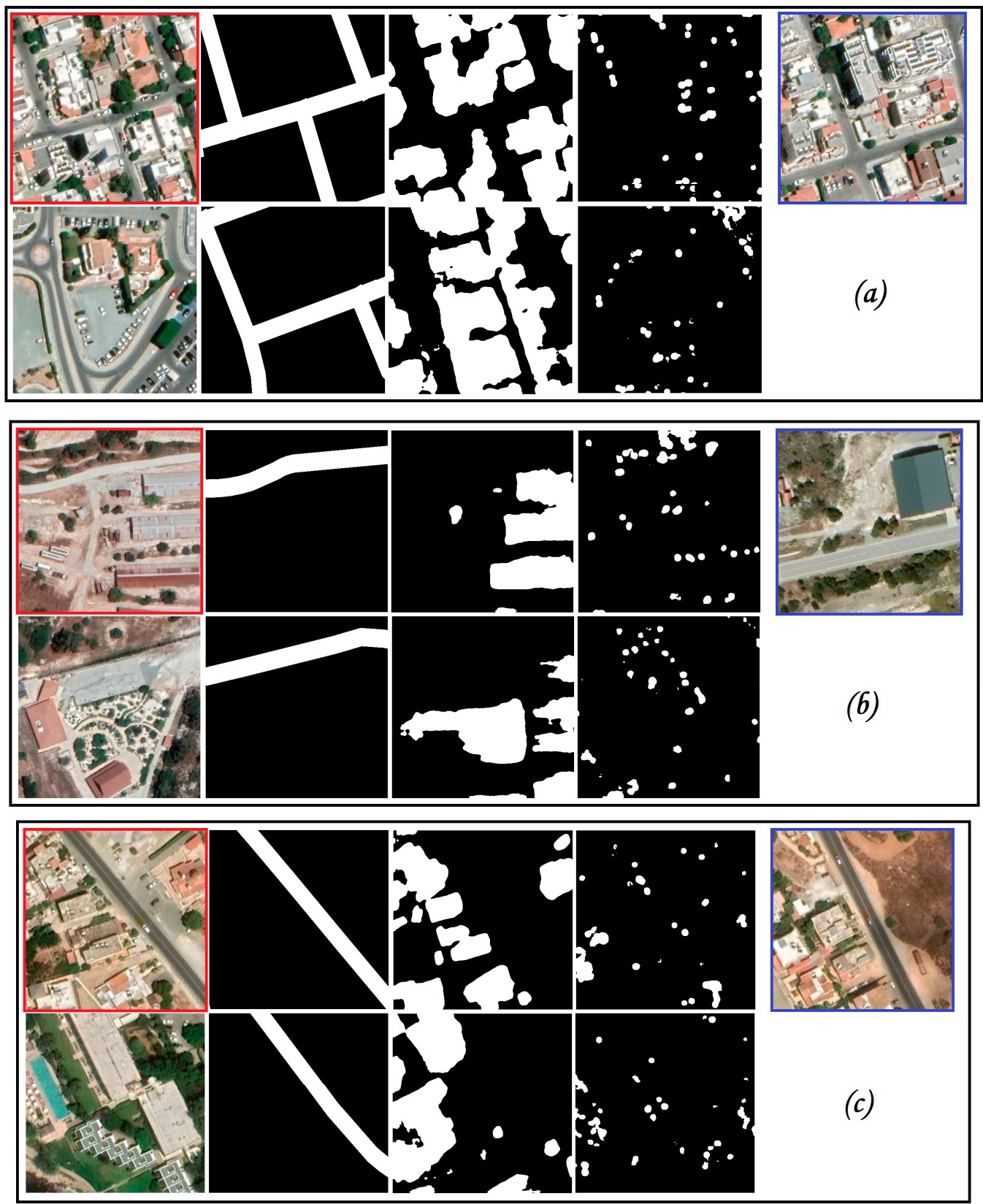

**Figure 6.** *Cont.*

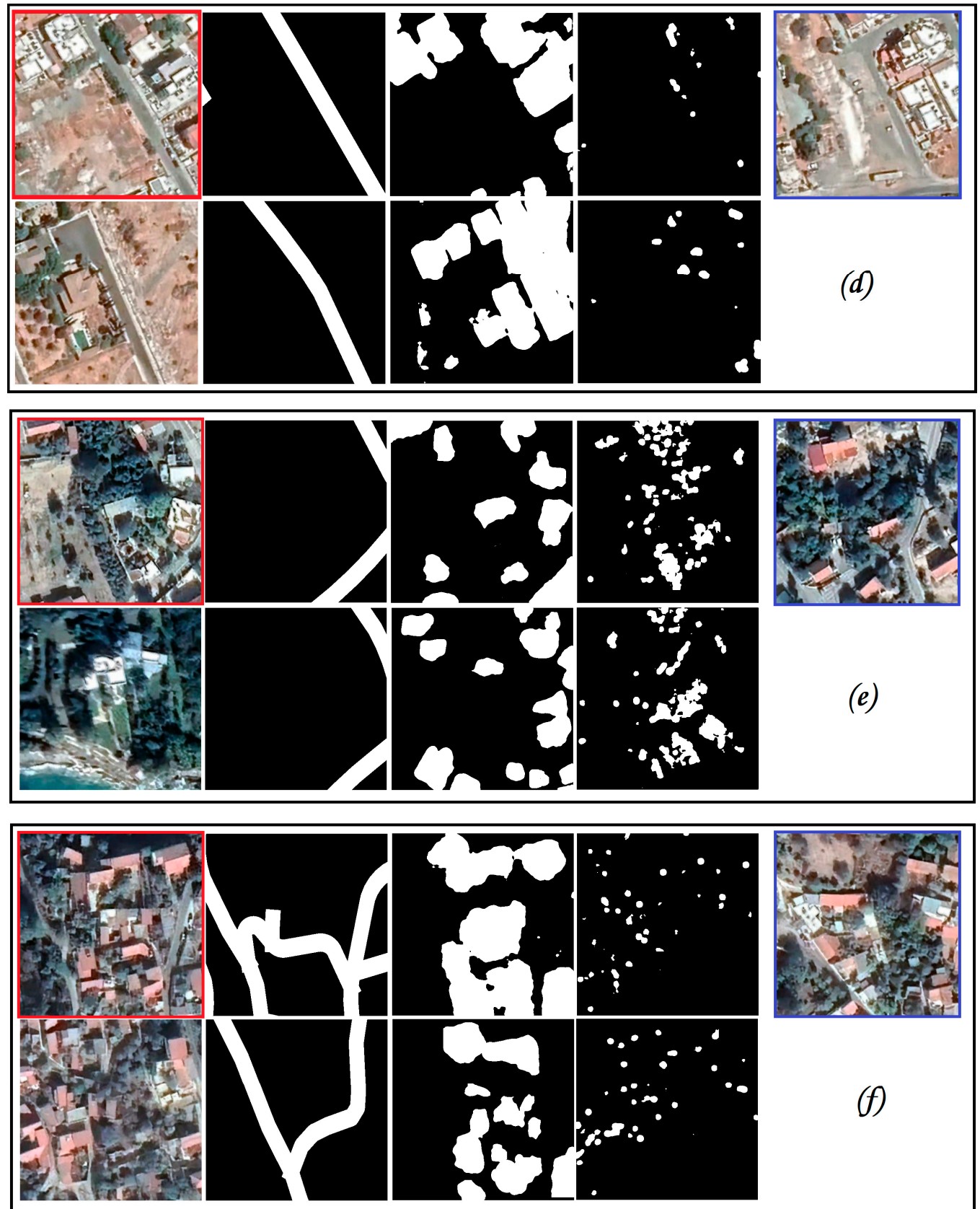

**Figure 6.** Six example queries, the representations of their underlying concepts, and the model output. Each query group (**a**–**f**) is visualized in two rows. The first row contains the query RGB (leftmost image

inside a red frame), the binary images of the concepts of the query image (roads, buildings, trees), and the output of our model (inside a blue frame); the second row of each group contains the best RGB match (returned by the RGB model alone) and the best matches for the 3 concepts (as indicated by the similarity models individually). The final output combines the similarities detected without fully adopting either of the individual concepts' matches. The fact that the output of our approach was a better match than the best RGB match alone, suggests that breaking up the similarity search task into individual concepts is beneficial.

## 4. Discussion

Breaking down the task of retrieving similar landscapes into domain-specific concepts is a strategy akin to how humans perceive the similarity between two optical satellite images: some concepts are dominant and tend to overthrow the prevalence of other concepts. For example, the tree arrangement might be of lesser importance than how large buildings are scattered in an image, and the texture of a large field might be unimportant when the layout of the road network and the layout of the buildings of two images match. This hierarchical concept cannot be exploited when a single model is used and no individual concepts are identified. Of course, this hierarchy can be modified according to the preferences of the user and the application requirements (see Section 2.3 on tuning the importance of the similarity aspects).

In this work, we demonstrate a proof-of-concept application of the proposed approach based on satellite images of the southern part of Cyprus. The approach could easily be scaled up without significant performance concerns to larger countries or even greater geographic areas, like continents. At such scales, inferred outcomes based on landscape similarity could be very impactful. For example, similar landscapes to a flooded area combined with additional information like altitude, slope, and soil type could indicate a high flood risk. A similar tree arrangement to populated areas that suffered from wildfires may indicate a substantial risk of fire spreading. Landscape similarity, and especially road network similarity, to areas with deadly car accidents might help policy makers understand the landscape conditions causing the accidents. This could then be correlated with speed limits in various locations or regions, to better shape safer and more convenient speed limits. We can also imagine applications in the real estate domain, where potential buyers create a draft drawing (could be a binary mask) of the ideal setting of their home (e.g., concerning trees, road network, buildings) and the best matches of landscapes with properties for sale are returned. An obvious match would be when a buyer is looking for a property in a cul-de-sac, while this info is not listed in the candidate properties' features. In the same context, landscape similarity inference (especially when reinforced with additional data) could support the buildup of knowledge regarding environmental issues.

### 4.1. Limitations

While the proposed approach achieved reasonable results, there are some important limitations regarding its implementation. First, the results were assessed based on human observation and not based on a quantitative performance metric. We plan to experiment with and propose meaningful metrics that could be used in this research area, to be able to set a baseline for comparisons of research efforts. Based on human observation and assessment alone, this process needs to be formalized by inviting various users, preferably experts in satellite imagery or landscape planning, to give scores to the landscape matching performed by different algorithms/approaches.

Furthermore, to determine the most similar landscape to a query, the proposed approach uses four DL models that are individually trained to perform different tasks. This inherently means that the errors of the models accumulate when they collectively contribute to the final decision. Since the Google Earth satellite images of Cyprus are often blurry, noisy, and of lower quality than other parts of the world (e.g., major, more densely populated regions on Earth), creating the datasets for the building and the tree masks to train the similarity models was hard and prone to errors. For similar reasons, the RGB

semantics' model was also prone to errors. Furthermore, the road network GIS layer also contained errors and this also contributed to the overall matching uncertainty.

*4.2. Future Work*

The problem of low-quality satellite images of regions on Earth that are not visited by satellites often enough to support the building of good-quality satellite imagery may be considered an opportunity for significant improvement in results' when the approach is applied to regions for which high-quality optical satellite imagery is available. In future work, we will apply our method to such areas and compare the results with the case of Cyprus.

A straight-forward next step would be to include other similarity concepts beyond buildings, trees, and roads, such as swimming pools and industrial units (which already play a role in the landscape matching process, see Figure 5), as well as railways, airports, sports outdoor fields, agricultural fields, parks, etc. The proposed approach does not explicitly account for any other concept besides the ones used to train it. However, due to using semantic similarity (RGB similarity model) as a contributing component, the proposed approach considers aspects of similarities that it was not explicitly trained on. This especially applies to large surfaces like water bodies (dams, lakes, and sea) that are generally matched by our approach due to the semantics similarity component. The tree/building/road masks in areas that depict water bodies are usually blank or contain very few trees or buildings (e.g., when the image belongs to a seaside city or town). This makes the semantic similarity model very important for carrying out the matching task. When the water body is too small and covers a very small fraction of the image, e.g., a normal sized swimming pool, it is mostly neglected by the approach because the predefined concepts dominate the decision making. To mitigate this behavior, the objects of interest should be predefined and used to train additional similarity models that contribute to the landscape matching decision.

Moreover, we intend to experiment with different weighting for each similarity concept, to understand the impact of different weights on the results. When embracing more similarity concepts, such as the ones suggested above, understanding the significance of each similarity concept would give insights into how humans define similarity in images, in particular landscapes visible from space.

Furthermore, our intention is also to experiment with images beyond optical satellite imagery. We aim to investigate landscape similarity when multispectral or hyperspectral satellite imagery is available, creating binary masks for certain bands beyond the optical/RGB at which important indicators for pollution/contamination, crop growth, and water stress, etc., are evident. This will allow us to match landscapes with similar characteristics and learn lessons about actions taken, e.g., to mitigate or avoid pollution, ensure food security, adapt to climate change, etc.

Finally, combining landscape similarity with multiple modalities (e.g., in situ field sensors and geospatial information) could be an exciting research field. Examples include better development of species distribution modeling, understanding the abundance of wildlife and biodiversity in regions with similar landscapes and micro-climates, deciding on the most appropriate speed limit on roads with certain characteristics, or valorizing properties more accurately, etc.

## 5. Conclusions

Identifying similarities in landscapes provides useful insights that can better shape policies and lead to better decisions by stakeholders in different application domains. Since the existing applications of similar landscape retrieval are limited by a moderate performance and the need for time-consuming and costly annotations, this paper proposed a method that involves splitting the similar landscape retrieval task into a set of smaller tasks that aim to identify individual concepts inherent to optical satellite images. Our approach relies on several models trained with unsupervised representation learning

(URL) on Google Earth images to identify these concepts. We demonstrated the efficacy of matching individual concepts for tackling the task of retrieving landscape(s) similar to a user-selected satellite image, with a proof-of-concept application of the proposed approach to the southern part of the island of Cyprus. Our results indicated the efficacy of breaking up the landscape similarity task into individual concepts closely related to remote sensing, instead of trying to capture all concepts and image semantics with a single model like a single RGB semantics model. Our method has certain limitations, but at the same time, there is potential for future work which could lead to significant new insights in this emerging research field.

**Author Contributions:** Conceptualization, S.K.; methodology, S.K.; software, S.K.; validation, S.K.; formal analysis, S.K. and A.K.; investigation, S.K.; resources, C.P.; data curation, C.P.; writing—original draft preparation, S.K.; writing—review and editing, A.K.; visualization, S.K.; supervision, A.K.; project administration, A.K.; funding acquisition, A.K. All authors have read and agreed to the published version of the manuscript.

**Funding:** This project received funding from the European Union's Horizon 2020 Research and Innovation Programme under Grant Agreement No. 739578 and the Government of the Republic of Cyprus through the Deputy Ministry of Research, Innovation and Digital Policy.

**Data Availability Statement:** No new data were created or analyzed in this study. Data sharing is not applicable to this article.

**Conflicts of Interest:** The authors declare no conflict of interest.

## Appendix A. The Data Generation Pseudocode for Training the Similarity Models

**Table A1.** The pseudocode for feeding the road network similarity model with pairs of similar road masks during training.

| **Input:** | Binary mask of the road network $m_1$ |
| --- | --- |
| **Output:** | A binary mask $m_2$ similar to $m_1$ ($m_2$ is a modification of $m_1$) |

```
m₂ := m1
### 10 Erosion/Dilation steps
for i = 1...10 do:
        kernelSize = randInteger(3,9)        # random kernel size between 3 and 9
        kernel = rectKernel(kernelSize)      # rectangle kernel
        if rand() < 0.5:
                m₂ = erode(m₂, kernel, iterations = 1)
        else:
                m₂ = dilate(m₂, kernel, iterations = 1)
### Randomly Translate
m₂ = RandomlyTranslate(m₂, 20)        # Randomly translate mask (±20 pixels)
### Rotate m₂ by 0°, 90°, 180° or 270° with probabilities [0.1, 0.3, 0.3, 0.3] respectively
angle = randomchoice([0, 90, 180, 270], p = [0.1, 0.3, 0.3, 0.3])
m₂ = rotate(m₂, angle)
return m₂
```

**Table A2.** The pseudocode for feeding the buildings' layout similarity model with pairs of similar building masks during training.

| **Input:** | Binary mask of the buildings' layout $m_1$ |
| --- | --- |
| **Output:** | A binary mask $m_2$ similar to $m_1$ ($m_2$ is a modification of $m_1$) |

```
m₂ := m1
### Randomly translate individual buildings in the image
bboxes = Mask2bbs(m₂)     ## buildings are contained in bboxes
for bb in bboxes:
        m₂ = RandomlyTranslate(bb, 20)     #Translate the content of each bb (±20 pixels)
```

**Table A2.** *Cont.*

```
### 10 Erosion/Dilation steps
for i = 1...10 do:
        kernelSize = randint(3,9)              # random kernel size between 3 and 9
        kernel = rectKernel(kernelSize)        # rectangle kernel
        if rand() < 0.5:
                m₂ = erode(m₂,kernel, iterations = 1)
        else:
                m₂ = dilate(m₂,kernel,iterations = 1)
### Rotate m₂ by 0°, 90°, 180° or 270° with probabilities [0.25, 0.25, 0.25, 0.25] respectively
angle = randomchoice([0, 90, 180, 270], p = [0.25, 0.25, 0.25, 0.25])
m₂ = rotate(m₂,angle)
return m₂
```

**Table A3.** The pseudocode of the augmentation process that feeds the RGB similarity model with pairs of similar RGBs during training.

| | |
|---|---|
| **Input:** | RGB image $i_1$ |
| **Output:** | RGB image $i_2$ similar to $i_1$ ($i_2$ is a modification of $i_1$) |

```
### Random Crop and Scale
### Final RGB size = 512 × 512 × 3
### Scale range = [0.2, 1.0], Aspect ratio = [0.75, 1.25]
i₂ = RandomResize(i₁, [0.2, 1.0], [0.75, 1.25])
i₂ = RandomCrop(i₂, [512, 512])
### Horizontal Flip with p = 0.5
i₂ = HFlip(i₂, 0.5)
### Apply Color Jitter (brightness= 0.8, contrast = 0.8, saturation = 0.8, hue = 0.8) with p = 0.75
i₂ = RandomColorJitter(i₂, [0.8, 0.8, 0.8, 0.8], 0.75)
### Convert to grayscale with p = 0.2
i₂ = Grayscale(i₂, 0.2)
return i₂
```

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
