# Peer review of "Scalable Retrieval of Similar Landscapes in Optical Satellite Imagery Using Unsupervised Representation Learning"

_remotesensing, doi:10.3390/rs16010142_

Round 1

Reviewer 1 Report

Comments and Suggestions for Authors

Generally, the topic is interesting and the manuscript is well written. However, there are some problems that need to be carefully addressed.

1. Quantitative analysis. The authors present some results for landscape similarity searches. However, this is not enought. What's the accuracy of the proposed method? The accuracy should be quantitatively presented.

2. Comparison with other models. There is no experiment showing that the proposed method is better than the existing methods.

3. In abstract, the authors claimed that: However, current applications of similar landscape retrieval are limited by moderate performance and the need for time-consuming and costly annotations. There should be analyses on how the proposed methods can address these problems in the paper.

Comments on the Quality of English Language

Minor editing of English language required.

Reviewer 2 Report

Comments and Suggestions for Authors

This paper presents an analysis on the scalable retrieval of similar landscapes in optical satellite imagery using unsupervised representation learning. The paper reads well and can be accepted after the below revisions:

1. Abstract: Too long, recommend to provide key aspects of the study.

2. Introduction: Line 30-50-> A long paragraph without any citation. Recommend to revise.

3. Figure 1: Same level of all sub figures are required.

4. Table 1: It can go to the appendix.

5. Table 2: Same comment as above.

6. Table 3: Same comment as above.

7. Figure 6: Sub-grouping is important.

Reviewer 3 Report

Comments and Suggestions for Authors

The paper presents a novel approach for scalable retrieval of similar landscapes in optical satellite imagery using unsupervised representation learning (URL). The proposed method is based on the idea of breaking down the complex task of landscape similarity retrieval into a set of smaller tasks that aim at identifying individual concepts inherent to satellite images. The authors demonstrate the efficacy of their approach on a proof-of-concept application using Google Earth images of the Republic of Cyprus. The paper addresses an important and timely problem in the field of remote sensing. The proposed method is based on a sound theoretical foundation and is well-motivated. The experimental results are promising and demonstrate the potential of the proposed approach. Overall, the paper presents a valuable contribution to the field of remote sensing. The proposed method is novel and has the potential to significantly improve the scalability and performance of landscape similarity retrieval. I recommend a minor revision.

1. Add more references in the introduction to support your statement.

2. all figures caption should be modified as it should be descriptive and to the point and if there are many sub-figures add a), b), ...etc.

Reviewer 4 Report

Comments and Suggestions for Authors

The paper is well organized. The abstract is informative and the introduction covers all relevant literature.

The references are up to date.

Hoever, 

1. The method in Section 2 in not adequatly supported by mathematics.

2. The figures captions in most are very large. This causes the reader to distract from the text.

3. Figure 5 and Figure 6 must be break down in smaller figures.

I suggest that the manuscript needs major revision.

Reviewer 5 Report

Comments and Suggestions for Authors

The article deals with a very interesting topic regarding the ‘Scalable Retrieval of Similar Landscapes in Optical Satellite Imagery Using Unsupervised Representation Learning’. Overall, it is a comprehensive article and the findings provided indicate that a great deal of effort was put in. Suggestions for improvements that could be performed to the manuscript before its publication are the following:

Abstract:

- It is suggested to avoid acronyms in this part of the manuscript in order to keep the Abstract simple yet informative. Those could be addressed in more detail throughout the following parts of the article.

- line 24: What do you mean by the expression 'individual concepts closely related to remote sensing'? What are those concepts and how are they related to remote sensing? It might be useful to elaborate more on this later on.

Introduction:

-line 86: Is there a particular reason why Cyprus was selected as the area of interest (because of the terrain, data availability, etc)? If so, it would be interesting to add those facts that made you select this specific area for your research implementation. In addition, would your results be likely to change if you chose a different study area or it would be irrelevant?

Moreover, your AOI selection contradicts the comments on lines 614-618.

Materials and Methods:

-lines 121-127: It is suggested to add some more relevant references at this point, in order to support what you mention.

lines 137-138: What about water surfaces/bodies? A relevant comment would be interesting. There is a mention of swimming pools in line 630, but it would make the concept more understandable if you comment about water bodies in general.

lines 209-2011: You mention that you selected to use Google Earth images for your image retrieval application because they provide high spatial resolution, they are relatively easy to acquire and they are frequently updated. The same goes for Sentinel-2 images that derive from the Sentinels repository. Therefore, it would be interesting to add a comment about the reasons for not selecting that option.

lines 235-236: Could any type of uncertainties/errors be inserted, by extracting a mask from A GIS road network layer? You mention something similar in line 618 (limitations) however, you do not propose a possible solution.

lines 245, 271, 285, etc: Figures' captions are too long. It is suggested to shorten the captions and refer to the figures in more detail throughout the text.

line 372: It might be better to refer to Milvus vector database at this point and not later on (next page). Or move part 'Vector Databases' in general before 'Similar Landscape Retrieval Process'.

Discussion:

- lines 585-587: Landscape similarity applications are very well explained. However, what are the facts that could support this note?

Limitations:

-Could the obstacle mentioned in the first paragraph be surpassed by using a more automated procedure (excluding as much as possible the human factor)?

Future Work:

-lines 623-628: The revisit frequency of each single SENTINEL-2 satellite is 10 days and the combined constellation revisit is 5 days with a spatial resolution of 10 m [bands: B2 (490 nm), B3 (560 nm), B4 (665 nm) and B8 (842 nm)].  In addition, SENTINEL-2 mission provides systematic coverage over all continental land surfaces (including inland waters) between latitudes 56° South and 82.8° North, all EU islands and all islands greater than 100 km2. Therefore, this could tackle the problem you mention.

-Overall, the 'future work' section is well structured.

Comments on the Quality of English Language

Minor editing of English is language required

Round 2

Reviewer 1 Report

Comments and Suggestions for Authors

I have no more questions.

Reviewer 4 Report

Comments and Suggestions for Authors

The authors have covered all requirements.